# Replication of population-level differences in auditory-motor synchronization ability in a Norwegian-speaking population

Guro S. Sjuls [1✉], Mila D. Vulchanova [1] & M. Florencia Assaneo [2]

The Speech-to-Speech Synchronization test is a powerful tool in assessing individuals' auditory-motor synchronization ability, namely the ability to synchronize one's own utterances to the rhythm of an external speech signal. Recent studies using the test have revealed that participants fall into two distinct groups—high synchronizers and low synchronizers— with significant differences in their neural (structural and functional) underpinnings and outcomes on several behavioral tasks. Therefore, it is critical to assess the universality of the population-level distribution (indicating two groups rather than a normal distribution) across populations of speakers. Here we demonstrate that the previous results replicate with a Norwegian-speaking population, indicating that the test is generalizable beyond previously tested populations of native English- and German-speakers.

[1] Language Acquisition and Language Processing Lab, Norwegian University of Science and Technology, Department of Language and Literature, Trondheim, Norway. [2] Institute of Neurobiology, National Autonomous University of Mexico, Santiago de Querétaro, México. ✉email: guro.sjuls@gmail.com

The Speech-to-Speech Synchronization (SSS) test is a short behavioral protocol for assessing human individuals' auditory-motor synchronization abilities[1]. The participants are instructed to align their own productions of a syllable (e.g., "ta") to a speech-proxy consisting of syllables presented at the average cross-language syllabic rate of 4.5 syllables/s[2,3]. Individual synchronization ability is then established by estimating the stability of the phase difference between the perceived and produced signals (Fig. 1). Remarkably, previous results have found population-level differences between the tested individuals, in that the ability to synchronize is bimodally distributed. In other words, a subgroup of the population (high auditory-motor synchronizers) spontaneously align their produced syllabic rate to the rate of the speech-proxy, while the other subgroup does not (low auditory-motor synchronizers). Furthermore, individual synchronization ability remains relatively constant across time (when tested 1 month apart), indicating a stable individual trait[1].

Several studies have replicated the bimodal distribution and established an association between performance on various cognitive and linguistically relevant tasks and membership in the high or low groups. For example, high synchronizers outperform low synchronizers in a statistical word-learning task[1,4], which could be supported by their selective activation of a frontoparietal brain network, in addition to a network of auditory and superior pre/motor regions which is similarly activated across high and low synchronizers[4]. Furthermore, speaking rhythmically entrains perception only for high synchronizers: for highs, syllables embedded in noise are better identified when presented at a specific phase of their speech-motor cycle, while, for low synchronizers, performance is not modulated by the motor phase[5]. With regards to the optimal range for temporal judgments (which is usually considered to be <10 Hz), high synchronizers have been found to have an extended optimal range, in the direction of faster rates[6]. Such differences in behavioral outcomes are likely supported by observed functional and structural neural differences between the groups[1,6,7]. Taking group membership into account has also led to the discovery of experimental outcomes that are not observed when merging all participants into one group[5–7], which speaks to the relevance of considering basic

inter-individual differences for accurate description of cognitive abilities and phenomena. As the protocol for establishing group membership is short and easily assessable[8], and the results predictive of various individual (behavioral and neural) differences of relevance to a range of studies, the test has the potential of being a reliable tool for future research.

However, although the bimodal distribution and characteristics related to group membership have been explored in several studies, the findings are limited to native English-[1,4,7–10] or German-[5,6,11] speaking individuals. While this may reflect common trends in psycholinguistic research[12,13], in order to confirm the generalizability of the previous results, and thus, the true relevance of the protocol, replications from a broader set of languages are required. Furthermore, although now replicated in several studies, the bimodal distribution of auditory-motor synchronization ability was initially an unexpected result, in that performance on cognitive and behavioral tests usually follows a normal curve. Positive, original and unexpected findings are more likely to be accepted for publication (in high impact journals), but are less likely to replicate[14–16]. Thus, the bimodal distribution of the SSS test should be carefully replicated, given the context of the ongoing replication crisis[17].

Replicating the SSS test with a Norwegian-speaking population provides information on the generalizability of the previous results. Norwegian is similar to English and German in aspects likely to affect perception and production of syllabic rhythms, as they are all stress-timed Germanic languages[18]. For languages like these, an approximately equal interval between two stressed syllables is expected, compared to every syllable being equal in duration (syllable-timed languages, e.g., Italian and French) or every mora being equal in duration (mora-timed languages, e.g., Japanese). Thus, a similar distributional pattern in auditory-motor synchronization ability as previously reported is hypothesized for native Norwegian-speakers. Unlike German and English, Norwegian is a bitonal language. Namely, it contains a prosodic feature with an opposition of two tones, whereby tone assignment is lexically determined and associated with the stressed syllable of a (bi-syllabic) word. If the hypothesis is confirmed, it indicates that synchronization ability is independent of this factor.

The hypothesis was tested by replicating the explicit accelerated version of the SSS test, where the participants are explicitly instructed to synchronize, while the syllabic rate is increased from 4.3. to 4.7 syllables/s, in steps of 0.1 syllables/s over the duration of the 1-min stimulus. In the current study, we firstly replicated the original SSS study, using the same syllabic stream (synthesized syllables consisting of English phonemes). Then, a second cohort of participants performed the same test, with a stimulus which phonologically is much closer to their native language (synthesized syllables consisting of Swedish phonemes), to assess the generalizability of the results (see "Methods" for more details).

For each cohort, the phase-locking value (PLV) between the perceived and produced signals was computed, to estimate individual synchronization ability. The distribution of PLVs across participants was assessed by (1) testing the uniformity of the distribution, (2) fitting three Gaussian mixture models, with one to three components, to the data, and evaluating the best fit based on the Akaike Information Criterion (AIC), and (3) fitting a Gaussian mixture model to the data with the number of components informed by the AIC. Individual group membership for each participant as high or low synchronizer was established based on which component their PLV fell under. In other words, two distributions were obtained, one for the cohort of participants tested with the original, English, stimulus, and one for the participants tested with the Swedish stimulus. Furthermore, these two distributions were compared. In addition, the distribution of

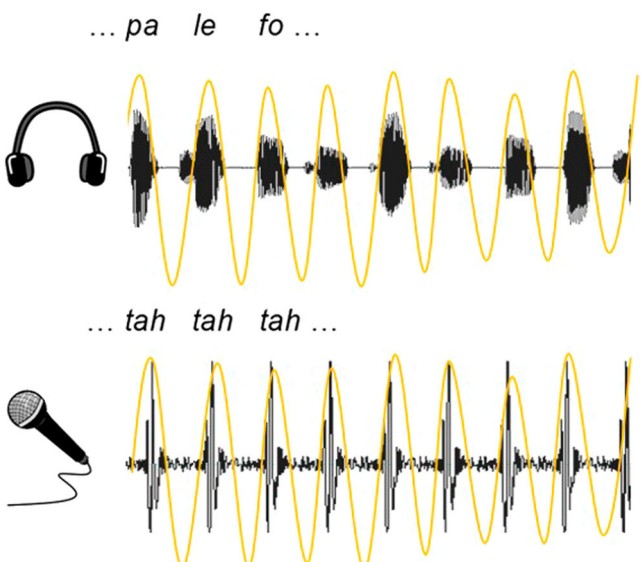

**Fig. 1 Example of the SSS test.** The upper panel represents the perceived, and the bottom panel the produced, speech signals. The yellow lines represent the envelope of the signals, bandpass-filtered between 3.5 and 5.5 Hz. Estimation of synchronization ability is quantified as the phase-locking value (PLV) between the filtered envelopes, for each participant.

the PLVs for both cohorts combined was assessed, as well as the same distribution stratified by sex. Lastly, group differences (i.e., low vs high synchronizers) for each cohort in terms of age, sex, years of education, years of musical training and level of musical experience were assessed. Data collection and analysis otherwise generally followed the established protocol[8] for the SSS test.

## Methods

**Participants**. Initially, a cohort of 72 participants completed the SSS test with the original (English) stimulus (23 male participants; mean age, 24 years; age range, 19–55 years). Eleven participants were removed because they spoke loudly instead of whispering, were silent for periods longer than 4 s, or because of extensive background noise, rendering the total number of included participants to $N = 61$ (19 male participants; mean age, 24 years; age range, 19–55 years). Following, a second cohort was tested, with a Swedish stimulus. A similar sample size to the first cohort was aimed for. As such, participants were tested until a sufficient number met the inclusion criteria for the SSS test, as described above ($N = 60$; 32 male participants; mean age, 30.5; age range, 20–64 years). Furthermore, a better sex balance was aimed for in the second cohort, given the results of the first cohort.

All participants were native Norwegian-speakers with self-reported normal hearing and no neurological deficits, recruited at the university campus. All participants provided written informed consent and the experimental protocol was approved by the Norwegian Center for Research Data. Participants in the English stimulus cohort received a gift card for their participation (as some additionally took part in another study), while participants in the Swedish stimulus cohort did not receive any compensation for their participation. The study was not preregistered.

**Questionnaire**. Each participant received a questionnaire, indicating sex, age, years of education, spoken languages, musical experience and whether a change in the SSS stimulus was detected. More specifically, they were asked whether they (1) perceived a change in the rhythm, (2) if they perceived at which time(s) it changed, and (3) if it was an increment or decrement of the syllabic rate. Musical experience was estimated based on (1) years of training with playing an instrument or singing, and (2) the level at which they trained (from self-taught, to professional level). Data on race or ethnicity was not collected.

**Stimuli**. For the first cohort, we used the same stimulus as in the original study[1], where 12 distinct syllables (unique consonant-vowel combinations) were semi-randomly concatenated into a 1-min syllable stream. There were no gaps between syllables and the only constraint in terms of the order was that the same syllable was not repeated consecutively. The syllable stream was synthesized using the MBROLA software[19] with the American Male Voice diphone database (us2) at 1600 Hz. The phonemes were equal in pitch (200 Hz) and pitch rise and fall (with the maximum at 50% of the phoneme). The duration of each phoneme was set to satisfy an increment in the syllabic rate from 4.3 to 4.7 Hz in steps of 0.1 Hz every 10 s over the duration of the syllabic stream.

For the second cohort, we used a stimulus synthesized in the same way as described above, but with the Swedish male diphone database (sw1). This allowed for generalization of the results to a language which is phonologically closer to the native language of the participants. Swedish was chosen as the MBROLA software does not contain a diphone database for Norwegian.

**SSS test**. The explicit accelerated version of the SSS test[1,8] has been conducted. Participants were seated in front of a computer

and were wearing over-ear Sony headphones (WH-1000XM4) and presented binaurally with the stimulus, at a mean air pressure of 75 dB. As the assessment had previously been tested online, as well as in a sound isolated booth, the current study allowed for testing participants outside of the lab in quiet environments. The test was carried out in PsychoPy software[20]. Summarizing, the participants underwent four main steps:

(1) Adjustment of volume until the participant could not hear their own whisper.
(2) Training step: the participants were primed with 10 s of the syllable "ta" at 4.5 syllables/s. Next, they were instructed to whisper "ta" with the same rate they just heard to familiarize themselves with the test before the experimental run.
(3) The synchronization task: The participants were instructed to synchronize to the perceived rhythm of the stimulus, by whispering the syllable "ta" (they were not informed about the increment in syllabic rate). Participants listened to the syllabic stream, while attempting to align their own productions to the perceived rhythm of the syllabic stream. The task lasted for 1 min and the participants fixated their gaze one a cross-hair on the computer screen throughout.

After the last step, participants filled out the questionnaire.

**Synchrony measurements**. Similarly to the original study[1], synchronization was measured by calculating the phase-locking value (PLV), using the formula:

$$\text{PLV} = \frac{1}{T} \left| \sum_{t=1}^{T} e^{i(\theta_1(t) - \theta_2(t))} \right|$$

where $t$ is the discretized time, $T$ is the total number of time points, and $\theta_1$ and $\theta_2$ are the phase of the envelope of the heard and the produced signal, respectively. The envelope of the signals was estimated as the absolute value of the Hilbert transform of the signal, envelopes were resampled at 100 Hz, filtered between 3.5 and 5.5 Hz, and their phases were extracted by means of the Hilbert transform. A bandpass filter [0, 3000] Hz was applied to the stimulus and the produced signals to eliminate potential background noise, as suggested in the protocol[8]. The PLV between the phases of the filtered envelopes of the perceived and produced signals was computed for windows of 5 s in length and with an overlap of 2 s. The results were averaged over all time windows, providing one PLV per participant. For more detail on the synchrony measurements, please refer to ref. [8].

In the original study, the PLV of two 1-min experimental runs was averaged. In the current study, only one run was completed for each participant. However, this should not introduce a substantial bias, as the PLVs from the two runs have been found to be highly correlated (Spearman rho, $r = 0.86$, $p = < 0.001$[1]).

**Spectral analysis**. For spectral decomposition of the produced speech signal, the discrete Fourier transform (DFT) of the envelope was computed, without any windowing. The power values were kept within a frequency window of [1, 10] Hz, and they were normalized to sum 1. The power estimates were averaged for high and low synchronizers.

**Statistical analysis**. The uniformity of the full distributions for each cohort was assessed with a one-sample Kolmogorov–Smirnov normality test, two-sided (alpha value = 0.05; the null-hypothesis (the distribution of PLVs is normally distributed) was rejected if $p < 0.05$). The results are reported as "$D = $ (degrees of freedom) KS test statistic", where $D$ is the maximum absolute difference between the empirical distribution function of the sample and the cumulative

distribution function of a normally distributed reference distribution (the Kolmogorov–Smirnov test statistic), given the degrees of freedom (which is equal to $N$). The KS test statistic provides a measure of effect size bounded between 0 and 1. Confidence intervals were obtained using a Monte Carlo simulation with 10,000 simulations, with a 95% confidence level.

High and low synchronizer groups were obtained by fitting a two-component Gaussian mixture distribution model to the PLV scores, using the diagonal covariance matrix and allowing for a maximum of 140 iterations[8,9]. The choice of number of components was informed by normalized Akaike's Information Criterion (AIC) values, which was estimated for three different models, one with one component, one with two components, and one with three components. The lowest AIC value was used to choose between the three possible models describing the data.

Group differences between highs and lows, in terms of age, years of education, years of musical training and level of musical expertise, were assessed with the non-parametric Mann–Whitney $U$ test, two-sided (the exact values are reported, at alpha value = 0.05). Effect sizes were calculated using two-sided rank-biserial correlations, which can be interpreted similarly to Spearman correlation coefficients.

Associations between group membership and sex was assessed with the non-parametric Fischer's exact test, two-sided (the exact values are reported, at alpha value = 0.05). Effect sizes were estimated by means of odd ratio, which can be read as the odds of a male participant being a high synchronizer, compared to a female participant being a high synchronizer, for the given cohort. Spearman's rank-order correlation, two-sided non-parametric correlation analysis (alpha value = 0.05) was computed to assess the correlation between PLVs and years of musical training.

Whether the PLVs obtained for each cohort come from different distributions was evaluated using a two-sample Kolmogorov–Smirnov test, two-sided (alpha value = 0.05; the null-hypothesis (the distribution of PLVs for each cohort comes from the same population) was rejected if $p < 0.05$. The results are reported as "$D$ = (degrees of freedom) KS test statistic", where $D$ is the maximum absolute difference between the empirical distribution functions of the two cohorts (the Kolmogorov–Smirnov test statistic), given the degrees of freedom (which is equal to $N$). The reported KS test statistic provides a measure of effect size bounded between 0 and 1. Confidence intervals were obtained using a Monte Carlo simulation with 10,000 simulations, with a 95% confidence level.

Additionally, Bayes Factor for independent samples was calculated to estimate the relative support for the null-hypothesis (the PLVs of the two cohorts come from the same distribution) compared to the alternative hypothesis (the PLVs of the two cohorts come from different distributions). Independent sample interference criteria were estimated using an adaptive quadrature method with tolerance set to 0.00001 and maximum iterations set to 2000. Prior distribution was estimated assuming unequal variance and Jeffreys priors, namely a non-informative prior distribution for parameter space. Bayes Factor was then obtained using Rouder's approach. Posterior mean = −0.029, 95% Credible Interval [−0.092, 0.035].

The distributions of PLVs for the two cohorts combined was also obtained, as well as distributions of the two cohorts combined stratified by sex. The uniformity of the distributions was assessed with a one-sample Kolmogorov–Smirnov normality test, two-sided (alpha value = 0.05). Confidence intervals were obtained using a Monte Carlo simulation with 10,000 simulations, with a 95% confidence level. High and low synchronizer groups for the distribution of the two cohorts combined were obtained by fitting a two-component Gaussian mixture distribution model to

the PLVs, using the diagonal covariance matrix and allowing for a maximum of 140 iterations[8,9]. The analysis was carried out in MATLAB R_2020.

**Reporting summary**. Further information on research design is available in the Nature Portfolio Reporting Summary linked to this article.

## Results

**Original (English) stimulus**. When computing the PLV between the perceived and produced signals for each participant ($N = 61$), the bimodal distribution of previous studies is replicated (Fig. 2A), as the synchronization ability of the tested individuals display two peaks. Statistical testing confirmed the non-uniformity of the distribution (one-sample Kolmogorov–Smirnov test, two-sided: $D(61) = 0.59$, $p < 0.001$, CI 95% [0.005, 0.008]), in support of our hypothesis. The bimodal nature of the distribution was further supported by fitting three Gaussian mixture models to the data, with one to three components (one, two or three peaks), and evaluating the best fit based on Akaike Information Criterion (AIC)[9]. For this cohort, the model with three components did not converge. The model with the lowest AIC, which is indicative of the best fit out of the two remaining models, was the one with two components (AIC$_1$ = −38.2217; AIC$_2$ = -53.9089). Based on the weight coefficients of the Gaussian mixture model[8] with two components, the cohort consisted of 25 high (mean PLV = 0.77, standard deviation = 0.045) and 36 low (mean = 0.48, standard deviation = 0.106) synchronizers.

The distribution of high and low synchronizers is slightly shifted in the direction of more low synchronizers, when compared to reports from previous studies. Namely, the weight coefficient for the low synchronizer component of the distribution is larger than the weight coefficient for high synchronizers. The weight coefficients (also called mixing parameters) of the GMM indicate how much of the respective component is in the resulting distribution, and equals to 1 for all components combined (here, weight coefficient lows = 0.62; weight coefficient highs = 0.38) (Fig. 2A). In addition, few participants ($n = 13$) correctly perceived a change in the stimulus, namely the increase in syllable rate from 4.3 to 4.7 Hz, which speaks for the automaticity of the synchronization process (Supplementary Fig. 2). In other words, participants did generally not consciously pick up on the acceleration of the stimulus. Similarly to previous reports, high synchronizers display less variability in their productions with a clearer peak around the frequency of the syllabic rate, as compared to lows, which is indicative of higher stability in their rhythmic productions (Fig. 2C).

The difference between high and low synchronizers is close to significant in terms of age, with low synchronizers displaying a larger standard deviation compared to highs (lows: median = 23, standard deviation = 7.6, highs: median = 23, standard deviation = 3.0; rank-biserial, $r = -0.23$; Mann–Whitney $U$ test, two-sided, $U = 312$, $p = 0.055$, 95% CI [22.73, 25.97]) (Supplementary Fig. 1A). There is no significant difference between groups in terms of years of education (lows: median = 15, standard deviation = 2.4, highs: median = 16, standard deviation = 2.2; rank-biserial correlation, $r = -0.10$; Mann–Whitney $U$ test, two-sided, $U = 412.5$, $p = 0.583$, 95% CI [15.45, 16.62]) (Supplementary Fig. 1B). There is, however, a significant association between sex and group membership, with a higher number of female participants in the low synchronizer group (lows: female participants $n = 31$, male participants $n = 5$, highs: female participants $n = 11$, male participants $n = 14$; Fischer's exact test, two-sided; $p < 0.001$; odds ratio = 7.89, 95% CI [2.30, 27.03]) (Supplementary Fig. 1E).

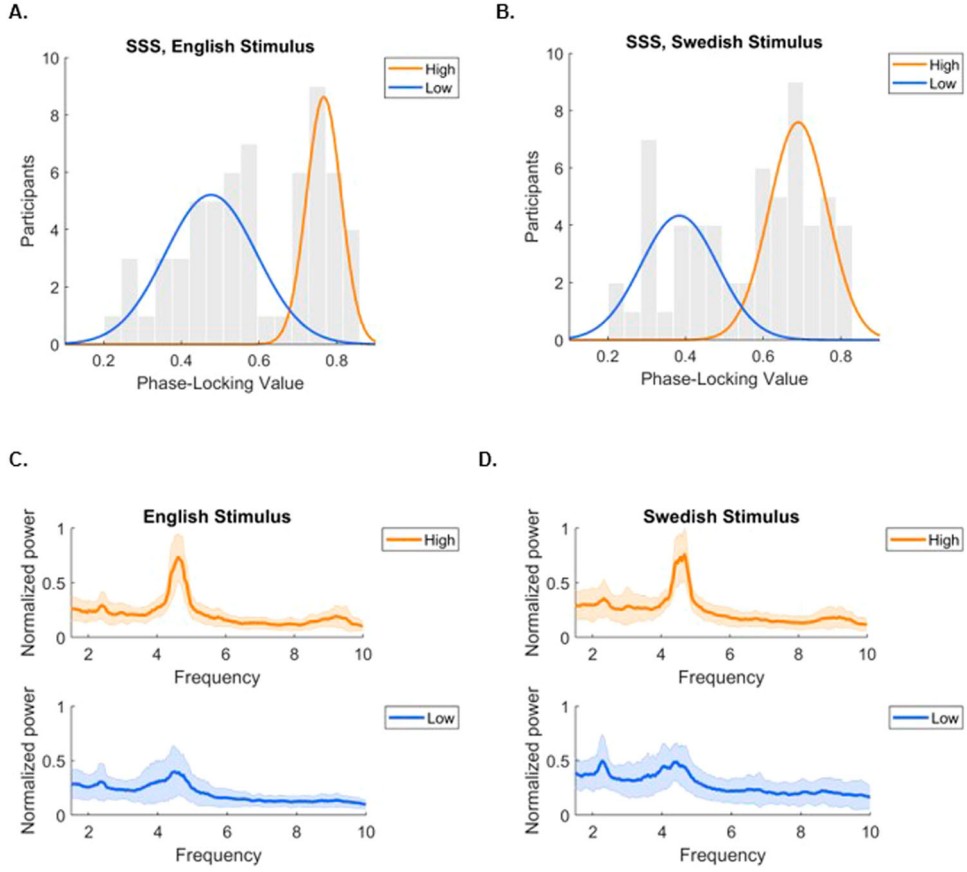

**Fig. 2 Results of the replication of the SSS-test in a population of native Norwegian speakers. A** PLV values for the English stimulus cohort, with distributions obtained by fitting the data with a two component Gaussian mixture model ($N = 61$): High synchronizers ($n = 25$), component weight coefficient = 0.38, mean = 0.77. Low synchronizers ($n = 36$), component weight coefficient = 0.62, mean = 0.48. **B** PLV values for Swedish stimulus cohort, with distributions obtained by fitting the data with a two component Gaussian mixture model ($N = 60$): High synchronizers ($n = 34$), component weight coefficient = 0.57, mean = 0.69. Low synchronizers ($n = 26$), component weight coefficient = 0.43, mean = 0.38. **C** Average spectra for the envelopes of the participants' produced speech, for the English stimulus cohort ($N = 61$). Standard deviation around the mean displayed in lighter color. **D** Average spectra for the envelopes of the participants' produced speech, for the Swedish stimulus cohort ($N = 60$). Standard deviation around the mean displayed in lighter color. For both cohorts, high synchronizers display greater stability in their rhythmic productions, as compared to lows.

There is a close to significant difference between the synchronization groups in terms of level of musical expertise (lows: mean = 1.7, standard deviation = 1.3, highs: mean = 2.6, standard deviation = 1.6; rank-biserial correlation, $r = 0.30$; Mann–Whitney $U$ test, two-sided, $U = 322.5$, $p = 0.055$, 95% CI [1.68, 2.45]), but not in overall years of musical training (lows: mean = 6.7, standard deviation = 10.2, highs: mean = 7.0, standard deviation = 5.8; rank-biserial correlation, $r = 0.02$; Mann–Whitney $U$ test, two-sided, $U = 342.5$, $p = 0.114$, 95% CI [4.61, 9.04]). We found no statistically significant evidence for a correlation between individual PLVs and years of musical training (Spearman rho, two-sided, $r(59) = 0.20$, $p = 0.118$, 95% CI [0.060, 0.438]). Only 12 participants had no experience with playing an instrument (Supplementary Fig. 1C, D).

**Swedish stimulus**. The PLV between the perceived and produced signals for each participant in the Swedish stimulus cohort ($N = 60$) also replicate the bimodal distribution of previous studies, as the synchronization ability of the tested individuals displays two peaks (Fig. 2B). The distribution is statistically non-uniform (one-sample Kolmogorov–Smirnov test, two-sided: $D(60) = 0.58$, $p < 0.001$, CI 95% [0.003, 006]). When evaluating the best fit out of three GMM models, with one to three components, the two-component model was the one with the lowest

AIC ($AIC_1 = -35.3429$; $AIC_2 = -46.0459$; $AIC_3 = -43.7129$). Based on the weight coefficients of the Gaussian mixture model[8] with two components, the sample consisted of 34 high (mean PLV = 0.69, standard deviation = 0.071, component weight coefficient = 0.57) and 26 low (mean PLV = 0.38, standard deviation = 0.096, component weight coefficient = 0.43) synchronizers. The distribution of high and low synchronizers is more in line with reports from previous studies, compared to the distribution for the English stimulus cohort, with a higher proportion of high synchronizers[1,8].

In addition, few participants ($n = 14$) correctly perceived a change in the stimulus, namely the increase in syllable rate from 4.3 to 4.7 Hz (Supplementary Fig. 4). Most participants did not, in other words, consciously pick up on the acceleration of the stimulus. High synchronizers display higher stability in their rhythmic productions, as indicated by a clearer peak around the frequency of the syllabic rate, compared to lows (Fig. 2D).

High and low synchronizers did not statistically differ in terms of age (lows: median = 27.5, standard deviation = 9.13, highs: median = 31, standard deviation = 7.5; rank-biserial correlation, $r = 0.13$; Mann–Whitney $U$ test, two-sided, $U = 333$, $p = 0.131$, 95% CI [28.37, 32,63]) or years of education (lows: median = 18, standard deviation = 2.8; highs: median = 18.5, standard deviation = 2.7; rank-biserial correlation, $r = 0.17$; Mann–Whitney $U$ test, two-sided, $U = 344.5$, $p = 0.143$, 95% CI [17.50, 18.94])

(Supplementary Fig. 3A, B). There is a close to significant difference between the groups in terms of level of musical expertise (lows: mean = 1.5 standard deviation = 1.3, highs: mean = 2.2, standard deviation = 1.4; rank-biserial correlation, $r = 0.24$; Mann–Whitney $U$ test, two-sided, $U = 320$, $p = 0.057$, 95% CI [1.56, 2.27]), and no significant difference between groups in overall years of musical training (lows: mean = 5.8, standard deviation = 6.9, highs: mean = 9.8, standard deviation = 9.1; rank-biserial correlation, $r = 0.23$; Mann–Whitney $U$ test, two-sided, $U = 323.5$, $p = 0.074$, 95% CI [5.93, 10.27]). Furthermore, individual PLVs and years of musical training are positively and significantly correlated (Spearman rho, two-sided, $r(58) = 0.32$, $p = 0.013$, 95% CI [0.062, 0.535]). Only 15 participants had no experience with playing an instrument (Supplementary Fig. 3C, D).

As a significant association between sex and group membership was observed in the English stimulus cohort, with a larger proportion of female participants, a more balanced sample was aimed for when the second, Swedish stimulus, cohort was recruited. Here, no significant association between sex and group membership was observed (lows: female participants $n = 14$, male participants $n = 14$, highs: female participants $n = 12$, male participants $n = 20$; Fischer's exact test, two-sided, $p = 0.435$; odds ratio = 1.56, 95% CI [0.56, 4.32]) (Supplementary Fig. 3E).

**Comparison of distributions**. Although the distribution of PLVs for each cohort varies somewhat at the descriptive level, e.g., in terms of number of high and low synchronizers, they were not statistically different from each other (two-sample Kolmogorov–Smirnov test, two-sided: $D(60, 61) = 0.16$, $p = 0.384$, CI 95% [0.337, 0.355]). There is substantial evidence in favor of the null-hypothesis, i.e., the two cohorts being similar (BFH0 = 4.8).

The distribution of PLVs for both cohorts combined ($N = 121$) also displays two peaks, and is statistically non-uniform (one-sample Kolmogorov–Smirnov test, two-sided, $D(121) = 0.58$, $p < 0.001$, CI 95% [0.014, 0.019]). The GMM model with the best fit was the one with two components ($AIC_1 = -76.7008$; $AIC_2 = -95.7426$; the model with three components did not converge). Based on the weight coefficients of the Gaussian mixture model[8] with two components, the combined cohorts consist of 47 high (component weight coefficient = 0.35, mean PLV = 0.75, standard deviation = 0.124) and 74 low (component weight coefficient = 0.65, mean PLV = 0.48, standard deviation = 0.051) synchronizers (Supplementary Fig. 5).

Because an association between sex and group membership was observed for the English stimulus cohort, but not for the Swedish stimulus cohort, the distribution of PLVs for both cohorts combined stratified by sex was also computed. Both distributions display two peaks and are statistically non-uniform (female participants: $n = 70$, one-sample Kolmogorov–Smirnov test, two-sided, $D(70) = 0.59$, $p < 0.001$, 95% CI [0.062, 0.072], male participants: $n = 51$, one-sample Kolmogorov–Smirnov test, two-sided, $D(51) = 0.59$, $p < 0.001$, 95% CI [0.014, 0.019]) (Supplementary Fig. 6).

**Discussion**
The results support the replicability and generalizability of the SSS test, speaking for population-level differences in auditory-motor synchronization ability as a potential universal trait, and confirming the primary hypothesis. Importantly, the distributions of the two cohorts, in which one was tested with an English and the other, a Swedish stimulus, were not statistically different. The joint distribution including both cohorts is statistically non-uniform, with the best fitting GMM model being the one with two

components, which further supports its bimodality. This speaks for the cross-language syllabic rate of 4.5 syllables per second as the most relevant aspect of the input, not the phonological nature of the stimulus or the native language of the speaker.

In the English stimulus cohort, a significant association between group membership and sex was observed, with more female participants in the low synchronizer group. Based on the current data, we cannot rule out the possibility that some sex differences might exist for Norwegian speakers when synchronizing to non-native sounding syllables. However, (1) a more balanced sample in the follow-up with the Swedish stimulus rendered no significant association between these variables, (2) when pooling together both cohorts, the distribution for each sex was significantly non-uniform, (3) previous reports, with higher sex balance, have not found this effect[1,5] (although, participants were not synchronizing to non-native syllables in these studies), and (4) most participants reported on being highly proficient speakers of English (English stimulus cohort, $N = 50$; Swedish stimulus cohort, $N = 54$), which makes it unlikely for this pattern to arise specifically for the English and not the Swedish stimulus. Taken together, this sex difference is therefore considered unlikely to drive the effect of the bimodal distribution in the English stimulus cohort. Still, if e.g., behavioral, or neural group differences are to be assessed, a balanced sample would be recommended.

In addition, previous studies have associated musical training with group membership, namely that high synchronizers are likely to have more years of musical training[1,11], which was not observed for either cohort. For the Swedish stimulus cohort, however, individual PLVs and years of musical training were significantly and positively correlated.

As compared to previous reports, the distribution for the English stimulus cohort was somewhat shifted, namely with few participants with a PLV of around 0.65 (Fig. 2A), whereas previous reports typically observe the division between groups at a PLV of around 0.5, with a larger proportion of participants in the high synchronizer group[1,8]. Thus, in the English stimulus cohort, the distribution displays a broader peak for low synchronizers, as compared to the relatively steep peak of high synchronizers, which differs somewhat from studies with English- or German-speaking individuals. This pattern could be attributable to the fact that participants were synchronizing to syllables of a non-native language which is phonologically different from their native language, which could be more difficult, leading to a higher proportion of low synchronizers. It is, however, likely not a general trait of Norwegian-speakers, as a pattern more in line with previous reports was observed when participants were synchronizing to a stimulus closer to their native language. However, the distribution of high and low synchronizers did not significantly differ between the two cohorts. As such, the specific pattern of PLVs for the English stimulus cohort might not be particularly relevant for the overall research question.

**Limitations**. The current study has only extended the results to a population fairly similar to the ones previously reported, in terms of overall similarities between the participants' first languages (with regards to rhythmicity), as well as the likelihood of the participants belonging to so-called WEIRD populations[21]. Still, the results indicate that previous findings replicate with native speakers of a bitonal language, indicating that the population-level distribution of synchronization ability is not affected by this factor.

Thus, building on the results presented here and in the other SSS studies, a wider range of languages should be assessed, preferably somewhat more distant than the three languages tested to date in terms of rhythmic class and other phonological

variables that might impact the results. As such, studies of syllable-timed and morae languages are encouraged. Importantly, in the context of finger-tapping experiments, which is another form of auditory-motor synchronization, utterances in stress-timed languages (i.e., English) are more readily synchronized to, as compared to utterances in a syllable-timed language (French), by both native English and French individuals[22]. Furthermore, the English L1 speakers were found to tap more regularly and at a higher rhythmical level to both the stress-timed and the syllable-timed utterances, as compared to French L1 speakers[22]. In other words, while group differences among individuals' auditory-motor synchronization ability have been found in three different populations of speakers, native language competence in stress-timed languages might have heightened these individuals' sensitivity for stress rhythms. This is also consistent with the language ranking positions established in Coupé et al.[23], whereby stress-timed intonation languages (such as e.g., English and German) are produced around the cross-linguistic mean rate of syllable per second, thus rendering them ideal for testing with the synchronization protocol. These languages thus appear more likely to display a more balanced bimodal distribution in synchronization ability. Future research needs to test this hypothesis.

## Conclusion

The primary hypothesis of the study was confirmed, as the bimodal distribution of auditory-motor synchronization ability was replicated with native Norwegian speaker. Furthermore, we show that the distribution is present both when participants are synchronizing to a language which is phonologically close to their native language, as well as a non-native language. As such, this indicates that synchronization ability is an individual trait which manifests when synchronizing to syllables presented at ~4.5 Hz, regardless of experience with the language. The results are promising in terms of the possibility of universal differences in an individual trait predictive of functional neural auditory-motor coupling, and behavioral outcomes, such as potential language-learning abilities.

## Data availability

As the data set contains individuals' voices, ethical restrictions with regards to sharing the raw data online or on a public server applies. However, they can be shared upon request send to the corresponding author. PLVs and the spectral content extracted from the speech signals to reconstruct Fig. 2, and numerical data to reconstruct Fig. 2 and Supplementary Figures are publicly available at https://doi.org/10.17605/OSF.IO/DWRZ7[24].

## Code availability

Code for running both versions of the SSS test online is available at https://app.gorilla.sc/openmaterials/290032[1], while code for running the SSS test and to analyze the data in-lab is available at https://zenodo.org/badge/latestdoi/407612860 (MATLAB version)) and at https://doi.org/10.5281/zenodo.6148008 (python analysis version)[25]. In the current study, the experiment was run in PsychoPy3 (code and stimuli available at https://doi.org/10.17605/OSF.IO/DWRZ7)[24]. Please see ref. [8] for the complete SSS test protocol. IBM SPSS Statistics version 27 and MATLAB version 2020 were used for data analyses.

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

## Acknowledgements

This work was supported by UNAM-DGAPA-PAPIIT IA200223 (M.F.A.), National Science Foundation (https://www.nsf.gov/) grant 2043717 (M.F.A.), and a strategic PhD University scholarship grant (Norwegian University of Science and Technology) (M.D.V. and G.S.S.). The funders had no role in study design, data collection and analysis, decision to publish or preparation of the manuscript. We thank Nora Nordseth Harvei, Sigurd Farstad Iversen, Alevtina Roshchina and Viktoriia Afoian for their help in the data collection.

## Author contributions

All authors contributed to the conceptualization of the study. G.S.S. was responsible for the data collection and performed the analyses and statistical testing under the supervision of M.F.A. G.S.S. wrote the first draft of the manuscript. All authors contributed to and approved the final manuscript.

## Funding

## Competing interests

The authors declare no competing interests.
