## [Peer Review File · Communications Psychology]

6th Apr 23

Dear Ms Sjuls,

Thank you for your patience during the peer-review process. Your manuscript titled "Replication in Support of Population-Level Differences in Auditory-Motor Synchronization Ability: Extending to a Norwegian-Speaking Population" has now been seen by 2 reviewers, whose comments are appended below. You will see that they find your work of potential interest. We would be very interested in considering a revised version for publication in the journal and inclusion in the Replication & Generalization Collection, provided the revision fully addresses these serious concerns.

We hope you will find the Reviewers' comments useful as you decide how to proceed. Should additional work allow you to address these criticisms, we would be happy to look at a substantially revised manuscript. If you choose to take up this option, please highlight all changes in the manuscript text file, and provide a detailed point-by-point reply to the reviewers.

The reviewers highlight a number of issues, of which we consider two critical enough to necessitate a replication of the present study in a Norwegian sample. The first issue are the gender differences, established in a sample that's comparatively smaller than your previous studies (e.g. 5). The second is the use of the American English voice from the MBROLA voices database. Although we understand that the replicated studies used the same voice, editorially we believe it would much strengthen your data to test whether the effect generalizes to other stimulus material. As the MBROLA database does not offer a Norwegian voice, you might want to consider whether you use another database, or if no suitable option is available, you might want to use another MBROLA voice from a Scandinavian language (see Reviewer #2). In your replication study, we ask you to follow SAGER guidelines (points 1-3, <https://www.nature.com/nature-portfolio/editorial-policies/ethics-and-biosecurity#sex-gender-sexual-orientation>).

The referees make a number of constructive suggestions regarding the presentation of the results. As you address these concerns, please bear in mind the journal's reporting standards, for example for statistics, which you can find in the checklist and template linked below (as well as on the journal webpage).

If the revision process takes significantly longer than five months, we will be happy to reconsider your paper at a later date, provided it still presents a significant contribution to the literature at that stage.

Please use the following link to submit your revised manuscript, point-by-point response to the Reviewers' comments with a list of your changes to the manuscript text (which should be in a separate document to any cover letter) and any completed checklist:

[Link redacted]

Please do not hesitate to contact me if you have any questions or would like to discuss the required revisions further. Thank you for the opportunity to review your work.

Best regards,

Marike Schiffer

Marike Schiffer, PhD
Chief Editor
Communications Psychology

EDITORIAL POLICIES AND FORMATTING

Editorial Policy: [Policy requirements](https://www.nature.com/documents/nr-editorial-policy-checklist.pdf) (Download the link to your computer as a PDF.)

Furthermore, please align your manuscript with our format requirements, which are summarized on the following checklist:

[Communications Psychology formatting checklist](https://www.nature.com/documents/commpsychol-style-formatting-checklist-article-rr.pdf)

and also in our style and formatting guide [Communications Psychology formatting guide](https://www.nature.com/documents/commpsychol-style-formatting-guide-accept.pdf) .

* **CODE AVAILABILITY:** All Communications Psychology manuscripts must include a section titled "Code Availability" at the end of the methods section. In the event of publication, we require that the custom analysis code supporting your conclusions is made available in a publicly accessible repository; please choose a repository that provides a DOI for the code; the link to the repository and the DOI must be included in the Code Availability statement. Publication as Supplementary Information will not suffice. We ask you to prepare and upload code at this stage, to avoid delays later on in the process.

* **DATA AVAILABILITY:**

All Communications Psychology research manuscripts must include a section titled "Data Availability" at the end of the Methods section or main text (if no Methods). More information on this policy, is

available at <http://www.nature.com/authors/policies/data/data-availability-statements-data-citations.pdf>.

At a minimum the Data availability statement must explain how the data can be obtained and whether there are any restrictions on data sharing. Communications Psychology strongly endorses open sharing of data. If you do make your data openly available, please include in the statement:

We recommend submitting the data to discipline-specific, community-recognized repositories, where possible and a list of recommended repositories is provided at <http://www.nature.com/sdata/policies/repositories>.

If a community resource is unavailable, data can be submitted to generalist repositories such as [figshare](https://figshare.com/) or [Dryad Digital Repository](http://datadryad.org/). Please provide a unique identifier for the data (for example a DOI or a permanent URL) in the data availability statement, if possible. If the repository does not provide identifiers, we encourage authors to supply the search terms that will return the data. For data that have been obtained from publicly available sources, please provide a URL and the specific data product name in the data availability statement. Data with a DOI should be further cited in the methods reference section.

REVIEWER EXPERTISE:

Reviewer #1 auditory neuroscience
Reviewer #2 auditory neuroscience

Reviewer #1 (Remarks to the Author):

The authors replicate an interesting result from a few years ago, namely that the ability to synchronize one's voice to an external rhythm is bimodal – people tend to be either good at this (high synchronizers) or not (low synchronizers). Here the result is replicated in Norwegian. The study is technically sound for the most part, and adds another language growing replication literature.

The main concern is the confound with sex/gender – the authors here find most of the high synchronizers are men, and most of the low synchroniser are women. It would be good to address this in some way:

1. The authors could stratify by gender, and see if there is bimodality in both men and women.
2. Include gender as a fixed effect or covariate in analyses and see if the results still hold.

3. Or perhaps some other method that the authors prefer and would be persuasive.

As the authors point out, previous studies have not shown this gender difference, but given that it has been found here, it is important to address it in some way.

Minor comments follow:

Lines 43-46: Maybe give examples of the associations you are talking about? Did the high synchronizers always perform better?

Line 48-49: Same comment – clarify what discovery are you talking about

Line 69: It is probably worth briefly defining stress-timed here as Comms Psych is a journal with broad focus.

Line 71: The authors hypothesise that the Norwegian pattern will replicate German and English, based on the fact that they are all in the same language family. This is reasonable. The meaning of the rest of the paragraph is unclear to me. It is odd to suggest a post-hoc explanation (bitonality) for an un-predicted null result in the introduction. More typically any speculation about a null result would occur in the Discussion, if the null result occurred. It is also unclear why hemispheric lateralization of tone perception in Norwegian is relevant (probably something to do with results of the original Nature Neuroscience paper, but this is not clear) Consider deleting that text, as it isn't really relevant given that the hypothesis was actually confirmed.

Line 91: The "Results" section seems to come out of nowhere. It would be better to include a paragraph at the end of the introduction briefly describing the methods.

Line 97: It is good practice to not report "p=.000". Either work out the p-value and express in scientific notation or use "p < .001"

Line 112: I know it is pedantic but rather than "similar in terms of age" authors should say the groups 'did not differ statistically'

Line 115: It is not clear how a Mann-Whitney U test is used to show that there was an association between male/female and high/low synchronizers. I recommend using a Fisher's exact test instead. <https://www.socscistatistics.com/tests/fisher/default2.aspx> (In fact, I ran this and it is p = .0003)

Line 123: Do label the colors on your plot.

Line 126: Do label the upper and lower panels on the plot.

Line 172: It still isn't clear how tonality of Norwegian would give rise to this specific pattern, with the wider distribution for low synchronizers. Could the authors try to re-write this in a more understandable way? Why would it only affect the low synchronizers?

Line 226: The authors note that underrepresentation of men did not introduce a bias in the original paper, but clearly in the present sample there is a large confound with sex/gender. It is probably best to delete this sentence.

Supplementary Materials: Include legends for the plots where appropriate

Reviewer #2 (Remarks to the Author):

Guro S. Sjuls and colleagues replicate and extend the established finding of a bimodal distribution of audio-motor synchronization abilities in Norwegian speakers. The approach and presentation of results are very clear and concise, and closely follow the previous studies.

The successful replication emphasizes the reliability of the synchronization test. As the authors state themselves in the discussion, Norwegian is rather close to the previously tested populations of German and English speakers, stressing more the replication than the novelty aspect of the presented work, which is nevertheless important. However, I wondered whether the stimuli should have been adapted to better match the properties of the native language of participants. Besides this point, I only have a few comments to enhance the clarity of the report.

My main comment concerns the stimuli: the authors use the same syllables as in previous studies, uttered by an American voice (l. 244 – 249).

Not adapting the stimuli might lead to differences between populations depending on the probability with which those syllables occur in their native language, and their proficiency in English. Do you think the observed shifts of the distributions with respect to previous studies could result from a lower familiarity with the syllables?

I think it would be important to test the synchronisation abilities with stimuli closest to participants' native language, or, if the authors disagree, this point should be explicitly addressed.

I could not find a direct report of the mean and standard deviations of the two Gaussian components of the phase locking values. For future studies it would be important to provide those measures as a comparison.

The caption of Figure 2, mentions the outcome of the Gaussian mixture model, but it is not clear to me how these parameters map on the mean and standard deviations of the two Gaussians depicted in the figure "component gaussian mixture model (Component 1, orange, high synchronizers; mixing proportion

(weight coefficient): 0.38, mean: 0.77. Component 2, blue, low synchronizers; mixing proportion (weight coefficient): 0.62, mean: 0.8)." Please explain how to interpret the weight and mean parameters of the Gaussian mixture models.

I also found it confusing that the term 'skewed' is used throughout the manuscript to describe the distributions, as Figure 2 shows perfectly symmetrical Gaussians. Please clarify which parameter of the distribution this skewness refers to.

Concerning the argument that higher musical training in the Norwegian population might have shifted the curves (l. 182 – 185): Do the individual phase locking values correlate with years of musical training (cf. Supplementary Figure 1)?

l. 185 – 187: "Lastly, related to the experimental set-up, the current study used the PLV from one run of the SSS test to estimate participants' synchronization ability, as the between-run correlation has been found to be substantial."

This sentence became clear only after I read the corresponding description in the Methods. Here, it reads as if the second run was recorded but discarded due to correlations. Please rephrase.

l. 295: Why were the power values normalized? Could this induce a relative shift across frequencies if the peak is not at the same frequency?

Response to Reviewers

Reviewer #1:

We highly appreciate you taking the time to read and make suggestions on our manuscript. We think the revised version has improved as a consequence of incorporating your suggestions. Thank you.

The main concern is the confound with sex/gender – the authors here find most of the high synchronizers are men, and most of the low synchroniser are women. It would be good to address this in some way:

1. The authors could stratify by gender, and see if there is bimodality in both men and women.
2. Include gender as a fixed effect or covariate in analyses and see if the results still hold.
3. Or perhaps some other method that the authors prefer and would be persuasive.

As the authors point out, previous studies have not shown this gender difference, but given that it has been found here, it is important to address it in some way.

Thank you for this comment. We have collected data from a second cohort of participants for the revised manuscript (who are tested with a stimulus that is phonologically closer to Norwegian), and for this cohort we strived for a more balanced sample in terms of sex. Here, sex and membership in the high or low synchronizer group was not significantly associated. To further estimate the likelihood of there being an association, given our original findings, we follow your advice and stratify the total sample (n=121) by sex. Distributions for both females and males are bimodal and are reported in the supplementary materials. We have added some points about this in the discussion.

Minor comments follow:

Lines 43-46: Maybe give examples of the associations you are talking about? Did the high synchronizers always perform better?

Line 48-49: Same comment – clarify what discovery are you talking about

Thank you, and yes, we see that this might be unclear. We have added a couple of sentences of the associations in this paragraph.

Line 69: It is probably worth briefly defining stress-timed here as Comms Psych is a journal with broad focus.

A brief description has been added.

Line 71: The authors hypothesise that the Norwegian pattern will replicate German and English, based on the fact that they are all in the same language family. This is reasonable. The meaning of the rest of the paragraph is unclear to me. It is odd to suggest a post-hoc explanation (bitonality) for an un-predicted null result in the introduction. More typically any speculation about a null result would occur in the Discussion, if the null result occurred. It is also unclear why hemispheric lateralization of tone perception in Norwegian is relevant (probably something to do with results of the original Nature Neuroscience paper, but this is not clear) Consider deleting that text, as it isn't really relevant given that the hypothesis was actually confirmed.

Thank you for highlighting this. The point of bringing up bitonality here was essentially to say that, if the previous results are replicated (if your hypothesis is confirmed) it would indicate that whether or not the native language of the speaker contains such a prosodic feature does not impact synchronization. Still, we see that the way it was originally written starts out at the wrong end of the argument, and we have therefore rephrased this section.

Line 91: The “Results” section seems to come out of nowhere. It would be better to include a paragraph at the end of the introduction briefly describing the methods.

A section has been added to introduce the results, thank you for the suggestion.

Line 97: It is good practice to not report “ $p=.000$ ”. Either work out the p-value and express in scientific notation or use “ $p < .001$ ”

Line 112: I know it is pedantic but rather than “similar in terms of age” authors should say the groups ‘did not differ statistically’

You are right, and this has now been changed throughout the manuscript.

Line 115: It is not clear how a Mann-Whitney U test is used to show that there was an association between male/female and high/low synchronizers. I recommend using a Fisher’s exact test instead. <https://www.socscistatistics.com/tests/fisher/default2.aspx> (In fact, I ran this and it is $p = .0003$)

This is a very good point, and we have now changed to the suggested test. However, there was a flaw in our original analysis/manuscript where a few of the participants who were supposed to be excluded were still in the dataset (accordingly, all of the related analyses have been run again), so the p-value is now $p < .001$. We used the same test to assess the association between group membership and male/female for the new cohort (where we aimed for a better sex balance), and we did not find an association here, as mentioned in response to the first comment.

Line 123: Do label the colors on your plot.

Line 126: Do label the upper and lower panels on the plot.

Supplementary Materials: Include legends for the plots where appropriate.

Thank you, these have been added.

Line 172: It still isn't clear how tonality of Norwegian would give rise to this specific pattern, with the wider distribution for low synchronizers. Could the authors try to re-write this in a more understandable way? Why would it only affect the low synchronizers?

Thank you for pointing this out. In the revised manuscript we have toned down this point, as the distribution of the new cohort is more in line with the studies we are replicating, and the distribution for the cohort synchronizing to the English stimulus and the cohort synchronizing to the Swedish stimulus do not statistically differ from each other. We have instead added a small point in which we suggest that synchronizing to syllables that are phonologically dissimilar to those of one's native language might, for some reason, be somewhat more difficult, leading to more people being classified as low synchronizers.

Line 226: The authors note that underrepresentation of men did not introduce a bias in the original paper, but clearly in the present sample there is a large confound with sex/gender. It is probably best to delete this sentence.

You are right, although that was the background for not prioritizing a more balanced sample to begin with. The sentence is nevertheless deleted.

Reviewer #2:

Thank you taking the time to read and provide feedback on our manuscript; it is greatly appreciated. We have incorporated most of your suggestions in the revised version.

My main comment concerns the stimuli: the authors use the same syllables as in previous studies, uttered by an American voice (l. 244 – 249).

Not adapting the stimuli might lead to differences between populations depending on the probability with which those syllables occur in their native language, and their proficiency in English. Do you think the observed shifts of the distributions with respect to previous studies could result from a lower familiarity with the syllables?

I think it would be important to test the synchronization abilities with stimuli closest to participants' native language, or, if the authors disagree, this point should be explicitly addressed.

Thank you for pointing this out. Originally, we did not think that this should matter too much for the replication of the results, given that most Norwegians are highly proficient speakers of English. We have however added a point in the discussion, referring to these points, stating that synchronizing to syllables that are phonologically different from one's native language might lead to a higher proportion of low synchronizers. More importantly, we have collected data from a second cohort of participants, who are synchronizing to syllables synthesized with a Swedish voice (as MBROLA, the program used for making the stimulus, does not have a Norwegian voice, and Swedish and Norwegian are phonologically similar). We also observed a bimodal distribution for this cohort, and the distributions of synchronization ability for the two cohorts did not statistically differ.

I could not find a direct report of the mean and standard deviations of the two Gaussian components of the phase locking values. For future studies it would be important to provide those measures as a comparison.

This has been added to the Results-section in the revised manuscript. Thank you for the suggestion.

The caption of Figure 2, mentions the outcome of the Gaussian mixture model, but it is not clear to me how these parameters map on the mean and standard deviations of the two Gaussians depicted in the figure "component gaussian mixture model (Component 1, orange, high synchronizers; mixing proportion (weight coefficient): 0.38, mean: 0.77. Component 2, blue, low synchronizers; mixing proportion (weight coefficient): 0.62, mean: 0.8)." Please explain how to interpret the weight and mean parameters of the Gaussian mixture models.

Thank you, and we see why this is not very intuitive, especially as there is a typo in the figure text: The mean for low synchronizers is $plv = 0.8$ in the original manuscript, which does not make much sense (as their plv typically is around 0.2-0.5). This has now been corrected. We have also added a couple of sentences to the Results-section, which we hope will clarify how to interpret the parameters of the Gaussian mixture models.

I also found it confusing that the term 'skewed' is used throughout the manuscript to describe the distributions, as Figure 2 shows perfectly symmetrical Gaussians. Please clarify which parameter of the distribution this skewness refers to.

We understand that the use of "skewed" can be confusing in this context. What we were referring to, is the tendency for our distribution (for the first cohort we tested, as described in the original manuscript) to have the lowest number of participants with a PLV of around 0.6. Namely, the point where the tails of the distributions for high and low synchronizers meet, lies around 0.6. While it in other studies (and in the second cohort we tested, as described in the revised manuscript) tends to be the lowest number of participants with a PLV of around 0.5. For our first cohort, this leads to the distribution of low synchronizers to be more "spread out", compared to previous reports. We have now described this in more detail and changed the term to "shifted" throughout. As per the comment above, we also hope that the now included description of the weight coefficients will make our point clearer.

Concerning the argument that higher musical training in the Norwegian population might have shifted the curves (l. 182 – 185): Do the individual phase locking values correlate with years of musical training (cf. Supplementary Figure 1)?

In the case of the original cohort, there is not a significant correlation, while there is one for the Swedish stimulus cohort. We have toned down this point, as the results are not the same for the two cohorts, which makes the results somewhat unclear. If anything, the Norwegian population, overall, has a fairly high number of years of training, but the level of this training varies. Based on our results, the level of training might be more relevant (we found a close to significant difference between groups for both cohorts for this variable).

I. 185 – 187: "Lastly, related to the experimental set-up, the current study used the PLV from one run of the SSS test to estimate participants' synchronization ability, as the between-run correlation has been found to be substantial."

This sentence became clear only after I read the corresponding description in the Methods. Here, it reads as if the second run was recorded but discarded due to correlations. Please rephrase.

Thank you for pointing this out to us. As the discussion has changed, given our additional results for the Swedish stimulus cohort, this sentence has been deleted.

I. 295: Why were the power values normalized? Could this induce a relative shift across frequencies if the peak is not at the same frequency?

These spectrums are used to show to which degree the population are synchronizing, given that the participants must be in synch with the specific frequency (which is in any case the same for all subjects) to get a high PLV. So, if the low synchronizers are keeping a different rate or being less isochronous it will in both cases imply that they are not able to synchronize their produced rate to the perceived rate. However, the broader peak in power for low synchronizers can have two explanations, namely that 1) each low synchronizer is keeping a different rate, or 2) all lows are less rhythmic (each have a broader peak). In any case, these power estimates highlight that one group of the population is overall more variable in their productions, compared to the other group, which is overall more consistent in their

productions. It should however be noted that the last sentence in the paragraph describing this analysis is removed (“normalized between 0 and 1”), as it was referring to the plots, which shows values between 0 and 1, not the normalization itself. “[...] normalized to sum 1” is correct.

30th Oct 23

Dear Ms Sjuls,

Your manuscript titled "Replication in Support of Population-Level Differences in Auditory-Motor Synchronization Ability: Extending to a Norwegian-Speaking Population" has now been seen by our reviewers, whose comments appear below. In light of their advice I am delighted to say that we are happy, in principle, to publish a suitably revised version in Communications Psychology under the open access CC BY license (Creative Commons Attribution v4.0 International License).

We therefore invite you to revise your paper one last time to address the remaining concerns of our reviewers and a list of editorial requests. At the same time we ask that you edit your manuscript to comply with our format requirements and to maximise the accessibility and therefore the impact of your work.

Please note that it may still be possible for your paper to be published before the end of 2023, but in order to do this we will need you to address these points as quickly as possible so that we can move forward with your paper.

EDITORIAL REQUESTS:

I'll highlight a few issues that are also listed in the "Editorial Requests Table" to make sure these aren't missed.

1) Reviewer #2 highlights the issue of missing Bayesian analysis in support of your claim of no difference between the groups tested on different stimulus material. It is indeed the case that this strong claim requires positive evidence for the null derived from Bayesian statistics or equivalence tests. In the absence of such evidence, you need to remove these claims and the discussion thereof from the manuscript. Please also note that marginally significant results may be stated, but not discussed.

2) Please clarify in the Data Availability statement what type of data (anonymized raw data, preprocessed data, data underlying the display items) can or cannot be deposited, what limitations exactly exist on data sharing for the anonymized raw data/preprocessed data, and to whom requests can be sent. Note the sharing requirements for the numerical data underlying the main analysis as reflected in the display items, both for the main manuscript and the SI.

3) You will need to provide us with updated versions of the Editorial Policy Checklist and Reporting Summary. In the previous version of the Reporting Summary, you noted that the analysis looks at "gender" differences, but the present version of the manuscript mentions "sex" and uses terminology associated with sex. Please ensure you provide precise details in both documents with regard to whether you collected and analysed data on sex or gender.

Please review all specific editorial comments and requests regarding your manuscript in the attached "Editorial Requests Table". Please outline your response to each request in the right hand column. Please upload the completed table with your manuscript files as a Related Manuscript file.

SUBMISSION INFORMATION:

In order to accept your paper, we require the files listed at the end of the Editorial Requests Table; the list of required files is also available at <https://www.nature.com/documents/commsj-file-checklist.pdf> .

OPEN ACCESS:

Communications Psychology is a fully open access journal. Articles are made freely accessible on publication under a [CC BY license](http://creativecommons.org/licenses/by/4.0) (Creative Commons Attribution 4.0 International License). This license allows maximum dissemination and re-use of open access materials and is preferred by many research funding bodies.

For further information about article processing charges, open access funding, and advice and support from Nature Research, please visit <https://www.nature.com/commspsychol/article-processing-charges>

At acceptance, you will be provided with instructions for completing this CC BY license on behalf of all authors. This grants us the necessary permissions to publish your paper. Additionally, you will be asked to declare that all required third party permissions have been obtained, and to provide billing information in order to pay the article-processing charge (APC).

* **DATA AVAILABILITY:**

[Link redacted]

Best regards,

Marike

Marike Schiffer, PhD
Chief Editor
Communications Psychology

REVIEWERS' COMMENTS:

Reviewer #1 (Remarks to the Author):

I am happy that the authors collected new data to address my concerns (thanks to them) and also happy with the manuscript, although it needs a proofread for typos:

"synchornizers"
"gaussian" (should be big G)
"weigh coefficient highs"

Also: "an equal interval between two stressed syllables is expected" <-- this should be changed to 'approximately' equal.

I have no further comments.

Reviewer #2 (Remarks to the Author):

The authors provide a thorough revision of the original manuscript, including data from a second cohort tested with stimuli voiced by a native speaker of Swedish, alleviating concerns about the previous results. Furthermore, this novel cohort allows to clarify the question about sex differences. The way the new cohort is included in the manuscript makes it clear that it was tested after analyzing the results of the first one, which I think is important for transparency. All comments I had previously have been addressed convincingly.

One last suggestion would be to add Bayes Factors, especially to the negative results if the claim to be made is that there is no difference between groups (rather than the absence of a significant difference).

Typo:

Results:

'The difference between high and low synchronizers is close to significant in terms age,'
-> in terms of age